# CD20 Expression as a Possible Novel Prognostic Marker in CLL: Application of EuroFlow Standardization Technique and Normalization Procedures in Flow Cytometric Expression Analysis

**DOI:** 10.3390/cancers14194917

**Published:** 2022-10-07

**Authors:** Anke Schilhabel, Peter Jonas Walter, Paula Cramer, Julia von Tresckow, Saskia Kohlscheen, Monika Szczepanowski, Anna Laqua, Kirsten Fischer, Barbara Eichhorst, Sebastian Böttcher, Christof Schneider, Eugen Tausch, Monika Brüggemann, Michael Kneba, Michael Hallek, Matthias Ritgen

**Affiliations:** 1Laboratory for Specialized Hematological Diagnostics, Medical Department II, University Hospital Schleswig-Holstein, 24105 Kiel, Germany; 2Department I of Internal Medicine, Center for Integrated Oncology Aachen Bonn Cologne Duesseldorf, German CLL Study Group, University Hospital Cologne, 50924 Cologne, Germany; 3Department of Hematology and Stem Cell Transplantation, West German Cancer Center, University Hospital Essen, University of Duisburg-Essen, 45147 Essen, Germany; 4Hematology, Oncology, Palliative Care, Clinic for Internal Medicine III, Rostock University Medical School, 18057 Rostock, Germany; 5Department III of Internal Medicine, Ulm University, 89081 Ulm, Germany

**Keywords:** CD20 expression, MRD, flow cytometry normalization

## Abstract

**Simple Summary:**

Up to 50% of patients with chronic lymphocytic leukemia relapse within four years after anti-CD20-directed treatment with a need for a subsequent treatment, which can potentially include anti-CD20 agents again. To retrospectively study the influence of CD20 expression in CLL patients receiving anti-CD20 directed therapy on therapy response, we designed and evaluated a bead-based normalization approach for the analysis of CD20 expression levels to reduce variation of flow cytometric measurements due to technical issues. Normalization significantly reduced the variability of median fluorescence intensities of fluorochrome-conjugated beads without artificially rendering the instruments’ performances, and longitudinal biological variations of marker expression based on MFI values could be robustly assessed. In a cross-trial comparison, strong MRD response correlated with the CD20 expression level before therapy started in patients receiving Ofatumumab, but not in patients receiving Obinutuzumab.

**Abstract:**

Background: CD20 expression is a controversial issue regarding response prediction to anti-CD20 therapy in chronic lymphocytic leukemia (CLL). Methods: Median fluorescence intensities (MFIs) of standard fluorescence beads from the daily calibration of flow cytometers according to EuroFlow protocols were used to establish a normalization approach to study CD20 expression on CLL cells. CD20 MFI was retrospectively assessed prior to and during treatment from flow cytometric measurements of peripheral blood in patients with different depths of molecular response in the four phase-II CLL2-BXX trials (BIG; BAG; BIO; BCG; *N* = 194) administering either Obinutuzumab or Ofatumumab in combination with targeted agents. Results: No significant difference was observed between the normalized and measured MFIs of CD19 and CD20 on CLL cells. During treatment, CD20 expression levels on CLL cells did not significantly differ between the four investigated different treatment schemes, but a strong molecular response to Ofatumumab seemed to correlate with higher CD20 expression prior to therapy. Conclusions: Standardized staining and instrument monitoring enable a robust assessment of longitudinal biological variations of marker expression based on MFI values. Obinutuzumab showed a higher proportion of patients with a strong MRD response independent from initial CD20 expression, whereas high pre-therapeutic CD20 expression levels seem to correlate with a profound response to Ofatumumab.

## 1. Introduction

The mechanisms of action and potential resistance against anti-CD20 type I and type II antibodies have been addressed in a number of studies during the development, engineering, and clinical testing of these drugs in different lymphoid malignancies (for references, see [1]). This has revealed, e.g., rapid antigenic modulation following treatment with anti-CD20 type I and type II agents in murine systems, human cell lines, and isolated chronic lymphocytic leukemia (CLL) cells [2,3,4,5]. The loss of CD20 on rituximab-treated cells in CLL and non-Hodgkin lymphoma (NHL) has been assigned to expression changes rather than the masking of CD20 by the anti-CD20 molecule [6]. There are only limited data on the long-term effects of anti-CD20 treatment on expression levels of this esteemed therapeutic target. Case reports have been published on reduced CD20 expression after rituximab treatment for NHL or CLL [7,8], also advising the assessment of CD20 expression prior to therapy decision. Current clinical guidelines allow repeated or salvage treatments with anti-CD20 antibodies in relapsed/refractory CLL [9]. Using the ratio of median fluorescence intensities (MFI) of CD20 on residual CLL cells and benign B-cells to the isotype control, Boettcher et al. observed a reduction of CD20 expression levels up to 180 days after the last therapy cycle in patients treated with FCR [10]. Data from real-world clinical care showed that up to 25% of CLL patients relapsed after frontline anti-CD20 chemoimmunotherapy (CIT) within 24 months [11]. CIT schemes normally include six 28-day cycles of anti-CD20 treatment, thus a limited number of patients would require the next treatment within the time frame in which decreased CD20 expression levels have been determined based on the flow cytometric MRD measurements [10].

Analyzing median fluorescence intensities (MFIs) by flow cytometry provides a measure for the presence (vs. absence) of expression of a given marker, as well as the amount of a targeted protein on the surface or even within the cells of different cell subsets. The technical complexity of flow cytometry and factors affecting the variability of measurements of samples at different time points and from different sources/patients have been addressed early through the usage of isotype controls. Traditionally, the fluorescence intensity of isotype controls has been used to distinguish between fluorescent positive and negative cells and evaluate non-specific binding in a sample. Different levels of the fluorochrome to protein ratio in monoclonal isotype controls, distinct immune globulin isotypes of the target antibody compared to the isotype control, as well as different autofluorescence of different cell subsets have been described as major drawbacks of isotype controls [12]. To overcome such issues, direct and indirect quantitation protocols, e.g., calibration curves from beads with a defined number of conjugated fluorochromes and stable ratios of fluorochromes conjugated to antibodies, were successfully introduced for quantitative analysis of marker expression [13]. More recently, the use of isotype controls as well as quantitation by calibration curves have been displaced by different approaches to achieve intra- and interlaboratory, as well as inter-instrument comparability of measurements. Besides using cell subsets for normalization [14], the application of uniform procedures for controlling the instrument performance and immune phenotypic staining of individual samples [15], rescaling instrument settings across different instruments and laboratories [16], or a combination of both [17], as well as automated processes for data clustering or alignment [18,19,20] have been appointed as main measures to reduce variance. Application of the EuroFlow procedure for instrument standardization in a multicenter study resulted in coefficients of MFI variation for the Rainbow beads ≤5%, compared to 11% to 48% for MFIs of stabilized cell subsets [15]. Notably, such standardization of instrument settings and regular quality control following the EuroFlow, as well as other similar standard operating procedures, have become highly accepted procedures in clinical flow cytometry laboratories.

Starting in 2015, the German CLL study group conducted several prospective CLL2-BXX studies (CLL2-BIG, -BAG, -BCG, -BIO) on sequential regimens using optional debulking with Bendamustine (B) followed by an induction therapy with anti-CD20 antibodies, either Obinutuzumab (G) or Ofatumumab (O) in combination with different B-cell-receptor signaling inhibitors (Ibrutinib (I), Idelalisib (C)) or the BCL2-inhibitor Venetoclax (A) and subsequent maintenance therapy combining the inhibitor and the anti-CD20 drug. All trials were open for newly diagnosed patients as well as patients with relapsed/refractory CLL [21,22,23,24]. The prognostic value of CD20 expression after CIT including anti-CD20 agents remains still a controversial issue, but CD20 levels seem to correlate with therapy outcomes [25,26,27,28]. To gain further insights and evaluate factors affecting weaker molecular responsiveness to such treatments, we retrospectively analyzed the CD20 expression levels in MRD samples of the above-mentioned four phase-II trials for patients with CLL. We sought possibilities to normalize MFI data that had been collected over several years on different flow cytometers. Here we report on an easy-to-implement normalization procedure with a proof of principle based on the MFI values of pre-established daily quality control standard beads. The effect of normalization was tested on the CD20 expression of CLL cells in patients with poor treatment response or relapsing after therapy in the trials. Our data also suggest that CD20 expression levels prior to therapy start may be useful for more effective therapy decisions.

## 2. Materials and Methods

### 2.1. Patients and Samples

For the patients enrolled in the phase-II CLL2-BXX trials CLL2-BIG (NCT02345863), CLL2-BAG (NCT02401503), CLL2-BCG (NCT02445131), and CLL2-BIO (NCT02689141), material for central MRD assessment was initially sent to 254 patients. Peripheral blood (PB) samples were collected in heparin before treatment (DX), at the end of induction treatment for final restaging (RE), and every 3 months during maintenance therapy (M). For a clinical description of the complete patient cohorts, we refer to [21,22,23,24]. Of the 254 patients with initial samples at baseline, 194 were evaluable for the present study based on the sample availability for the applied MRD scoring. Clinical and genetic data for the selected patients in the MRD response groups are summarized in Appendix A.

### 2.2. Molecular Response Scoring

Therapy response was categorized for patients based on the minimal residual disease (MRD) values at RE and the following MRD course during maintenance therapy. The response groups included patients with deep (undetectable MRD (uMRD) defined as MRD negativity even when detectable MRD levels are below 10^−4^), intermediate, limited MRD response, and non-responders, according to Table 1.

### 2.3. Flow Cytometric MRD Assessment

MRD was measured using a 4-color panel similar (4C MRD) to the one described by Rawstron et al. [29]. Instruments were operated under EuroFlow settings using fluorochrome-conjugated beads (8-peak Rainbow calibration particles (Thermofisher Scientific, Bremen, Germany) or CS&T beads (BD Biosciences, Heidelberg, Germany) weekly or daily to adjust the PMT voltages for the individual channels to reach the target MFI values [15] and underwent additional daily monitoring of the instrument performance. To study changes in CD19 and CD20 expression levels, the MFIs of both markers of CLL cell populations were normalized using a factor derived from the daily quality control, as described in the results section.

The normalization factor NF was calculated for each fluorochrome as the ratio of the measured MFI to the target MFI of the 7th peak of the Rainbow calibration beads (Equation (1)):NF = measured MFI peak 7/target MFI peak 7.(1)

Normalized MFIs were calculated by dividing the measured MFI by NF (Equation (2)):normalized MFI = measured MFI/NF.(2)

### 2.4. Determination of Flow Cytometric Measurement Variability

Variance in flow cytometric measurements was determined from a 5 × 3 × 3 experiment, testing the effect of the donor (biological variability), instrument (technical variability), and day (environmental variability) on measured fluorescence intensities. PB was collected from 5 healthy donors on 3 non-consecutive days. The samples were stained with the 4C MRD panel and analyzed on three different instruments. The antibody binding capacity of the PE-tagged targets CD56, CD22, CD38, and CD79b was determined from the consecutive measurements of three of the five donors using Quantibrite PE beads (BD BioSciences, Heidelberg, Germany) according to the manufacturer’s instructions. The MFIs for different populations were used to evaluate the effect of the selected parameter on overall variability using the eta squared value from ANOVA. F-statistics were determined to test models using only the donor as a factor against models using all factors.

### 2.5. Statistical Analyses

Data mining and statistical analysis were performed using R 4.0.3 (Vienna, Austria) and Rstudio 1.4.1103 (Boston, MA, USA) [30,31]. ANOVA and post-hoc tests were used from the main R package, the Kruskal–Wallis and Dunn’s test were performed to test the statistical significance of differences between groups, and the pairwise t-test was further used on significant groups for the comparison of normalized and measured MFIs. A mixed principal component analysis was performed using the package pcamixdata [32].

## 3. Results

The aim of this project was to evaluate the CD20 expression levels of residual CLL cells in those patients who did not respond to the treatments in the trials (NR), compared to patients who showed variable degrees of response on the molecular level (deep, intermediate, or limited response; see Table 1 for molecular response categories). As progression of MRD was observed at different levels and with different kinetics, patients who initially showed a good MRD response but progressing MRD did not reach 10^−4^ reduction compared to therapy start were scored differently (intermediate, IR) than patients whose MRD levels were roughly at 10^−2^ within the observation time (limited, LR) (Figure 1). From the initial number of 254 patients of the CLL2-BXX trials with samples at baseline, 60 patients were dropped from the dataset due to no or too few on-therapy samples or too low CLL counts at the time point of screening. Therefore, only 48% of patients in the CLL2-BCG trial (*N* = 23), but ~90% of patients from the described clinical cohorts of the other three CLL2-BXX trials could be included in the dataset (BAG: *N* = 57; BIG: *N* = 56; BIO: *N* = 58). Although slightly variable with respect to the cytogenetic parameter for the selected patients, only minor deviations from the complete clinical cohorts of the trials were observed [21,22,23,24]. Based on the defined cut-offs for MRD response, the majority of the patients (132, 68.0%) were in the MRD deep response group (uMRD), 14 patients fell in the non-responder category (7.2%), 23 showed a limited response roughly reaching MRD values < 10^−2^ (11.8%), and 25 had an intermediate MRD response but did not reach MRD negativity below 10^−4^ (12.9%) within 1.5 years maintenance therapy. Overall, 54% (*N* = 105) of the patients qualified for this study were treatment-naïve (TN) and 46% (*N* = 89) had relapsed/refractory disease (RR) (Appendix A).

MRD samples in the four trials were collected through a time span of 63 months and analyzed on different flow cytometers. Although instruments were standardized according to the EuroFlow standard operating procedure [15], the need for a normalization approach to reduce the influence of technical variation caused by the different instruments over time was tested.

### 3.1. Longitudinal Variability of Fluorescence Intensities of Fluorochrome-Conjugated Beads

Although CD3 expression levels on T cells can have a variability of approximately 40% [33], we tested the potential of measured CD3 MFI on T cells or CD3-negative cellular subsets (B cells) for normalization purposes. In addition to some expected variability, we noted a possible seasonal or environment-related trend in the MFI of the CD3 expression levels of T cells measured over an 8-year period, as well as similar fluctuations in the MFIs of the fluorochrome beads passing the target acceptance limits for daily quality control and instrument standardization (8-month period exemplarily displayed in Figure 2). Concordant changes were also recorded for the other peaks, covering the MFI range of the beads.

Comparison of the quality control MFIs measured on two CantoII and two Lyric instruments during an 8-month period in which all instruments operated simultaneously (Figure 2) revealed longitudinal variations with increasing or decreasing trends (e.g., CIIB, LB). Additionally, steep MFI increases or decreases (CantoII instruments), which did not necessarily correlate between fluorochromes (e.g., CIIA FITC and APC), were also observed. Whereas global environmental conditions can be neglected as sources of inter-instrument variation under these circumstances, the observed steps could only partially be assigned to changes in bead lots or maintenance. Such within-range but pronounced MFI changes increase the variability of MFI measurements in instrument standardization and could potentially contribute to a distortion of longitudinally measured values.

### 3.2. Bead-Based Normalization for Cantoii Data—Proof of Principle across MFI Range

We used a correction factor (Equation (1) in Material and Methods) of the ratio of the actually measured MFI to the target MFI to test: (1) whether the fluorescence changes are linearly correlated across the beads’ fluorescence intensity range, (2) if the fluctuations can be normalized using this ratio, and finally, (3) if such correction would influence results of expression analysis. Instant shifts can be compensated by applying the normalization factor (Figure 3, exemplarily shown for peak 2 of one instrument, Appendix A) across the MFI range of the quality control beads. The residual variability was reduced without distorting the longitudinal course, and, with exception of PE for the CantoIIA, CVs across the MFI range of the beads were reduced from ~1.8 fold for peak 2 up to ~8-fold for peak 6 across the different fluorochromes (Appendix A).

### 3.3. Sources of Variance in Flow Cytometric Measurements

To assess sources of variance in flow cytometric measurements of biological specimens, a 5 × 3 × 3 experiment was performed measuring the fluorescence intensities of markers for different cell populations in blood from five healthy donors on three non-consecutive days (within a 2-week time period) on three different instruments using the 4C-MRD panel that was also used for the CLL2-trial samples. As a control, the antibody-binding capacities of the PE-tagged targets (CD56 on NK cells (CD3-), CD22, CD38, and CD79b on B cells) were determined using a standard curve derived from Quantibrite-PE beads for the first three donors. Antibody-binding capacities (ABC) for CD22 (14,492 ± 1442) and CD79b (9697 ± 2097) were in concordance with previously reported values from healthy volunteers’ blood [34]. CD56 expression values on NK cells and CD38 expression values on B cells have not been found in the literature, as the used coupled fluorochromes were not PE and only MFIs were reported. Thus, MFIs were used in a linear regression analysis without applying a normalization factor to determine the effect size of the tested parameter on the overall variance from a larger donor pool.

The effect size η^2^ reported from the ANOVA indicates that the donor has the highest impact on the observed variability (Table 2). Across the different targets, up to 83% (mean 55.2%) of the measurement variability is caused by inter-donor variations. On average, only ~8% of the variance is related to the instrument, implying that the technical variability caused by measurements on different instruments is neglectable when operating under standardized conditions. Yet, on average, 28.5% of the overall variance is related to the time (day) of the measurement, which still includes several factors (unresolved in our experimental setup) such as intra-donor variation, variations caused by the used antibody batch, or the operator performing the staining or data evaluation.

### 3.4. CD20 Expression before Therapy—Applying Normalization

Despite the low impact of the technical variance on the measurements, the normalization factor was applied to data recorded with the MRD assessments of the CLL study samples. Although the normalized MFIs for CD19 and CD20 showed significant differences for the median values compared to the measured MFIs (*p* < 0.02), only a few samples showed minor deviations from the correlation (Figure 4). To observe possible distortion of MFI differences, normalized and measured values were used throughout all subsequent analyses (only normalized values are shown in the results section). *p*-values from pairwise t-tests also differed only slightly between normalized and measured values, and no changes in the significance of group differences were observed using either normalized or measured values of CD19 or CD20.

Overall, pairwise Kruskal–Wallis tests supported CD20 MFI differences for the different factors *IGHV* mutation (*p* = 1.8 × 10^−5^), *TP53* mutation (*p* = 0.016), trisomy 12 (*p* = 2.8 × 10^−5^), and response groups (Figure 5), but not for the factors *NOTCH1* mutation, gender, or the therapy strata. For the genetic factors, the significance levels vanished when expression levels were analyzed between the different MRD response groups.

Because of sufficient patient numbers, the group of the uMRD responders was divided according to their treatment. Surprisingly, this showed elevated CD20 MFI levels of the uMRD responders in the BIO trial with ibrutinib and Ofatumumab (mean MFI 6107 ± 5212) compared to the uMRD responders in the other trials before starting therapy (mean MFI of other uMRD groups 2205 ± 2079) (Figure 5). Patients in the response subgroups differed in age with groups with a mean age of ~60 years (uMRD-BAG, uMRD-BIO, and LR) and >65 years (uMRD-BCG, uMRD-BIG, IR, NR), but no correlation between age and CD20 expression was observed. The proportion of patients with Binet stage B or C at therapy start was increased in response groups IR, LR, and NR (78–88%) compared to the response groups achieving uMRD (53–75%). More patients in the younger uMRD groups were treatment-naïve, whereas the older uMRD groups and the groups with varying responses showed a more balanced proportion of patients in first-line and relapsed/refractory treatment (Appendix A). CD20 expressions in all treatment-naive patients were statistically not significantly different than for patients with replased/refractory CLL (*p* = 0.84; Appendix A). Even though the uMRD responders of the CLL2-BIO trial showed elevated CD20 MFI values compared to the other response groups, no statistically significant difference was observed for treatment-naïve (TN) and relapsed/refractory (RR) patients in this group (mean MFI_TN_= 6387 ± 5768; mean MFI_RR_ = 5686 ± 4741; *p* = 0.85; Dunn’s test), as well as for any of the other within-group comparisons (Appendix A).

Concerning other prognostic factors, on average, only 20% of patients in the uMRD groups harbored *TP53* mutations compared to 40.6% in the IR/LR/NR groups. Such a difference was not observed for the second prognostic factor, the status of *IGHV* mutation (average 63% uMRD vs. 76.5% IR/LR/NR not mutated *IGHV*; *p* > 0.05). *NOTCH1* mutations, known to influence CD20 expression [35,36], were present in 19% and 15.6% of patients of the uMRD and IR/LR/NR groups, respectively. However, trisomy 12, which has already been related to elevated CD20 expression [26], showed a clear shift in positive cases for the uMRD-BIO group (40%) compared to the other groups (on average ~15.4%).

To identify possible factors provoking the CD20 expression differences in patients in the BIO trial compared to the other trials, a mixed principal component analysis was performed using the CD20 expression levels, age, Binet stage, *IGHV* mutation, trisomy 12, and therapy strata as factors. Patients in different groups did not cluster before and after component rotation, although rotation clearly improved the loading of the factors, especially for age in component 3. Yet, patients in the BIO uMRD group are distributed in the same manner on the observation map of the PCA as patients in other response groups (Figure 6).

### 3.5. CD20 Expression under Anti-CD20 Maintenance Therapy

MRD samples at RE were available for all patients categorized in the response groups IR/LR/NR. Based on the indicated time of last anti-CD20 treatment after therapy started, samples from patients were grouped to allow the sample set to be as complete as possible for defined time intervals (9–12 months, 15–18 months, and >18 months) of maintenance therapy duration. Samples analyzed with a different instrument generation were excluded from the current comparison, but ~97%, 83%, and 70% of samples were still available for the timepoints ~2 months, ~9 months, and ~15 months after the end of induction treatment (final restaging: RE, maintenance cycle 3 or 5: M3, M5), respectively. CD20 expression seems to be highly variable in the response group LR within the observation time points (Appendix A). Compared to the level at therapy start, the MFIs at RE for CD20 and CD19 showed a 16- to 25-fold decrease and a 1.8- to 2.6-fold decrease, respectively. A trend for recovery of CD20 expression during the course of maintenance was only observed for a few patients, mainly in the NR group. For other patients, the degree of reduced CD20 expression did not change during the course of maintenance, and also, CD19 expression seems to recover only gradually to a 1.3- to 2.1-fold decrease. However, multifactorial ANOVA indicated differences between patients in response categories for CD20 expression (*p* = 0.0129), and the pairwise t-test using Bonferroni adjustment confirmed the differences in the responder groups IR and LR (*p* = 0.014) using combined maintenance-therapy samples (Appendix A).

A subset analysis was performed using only samples from the CLL2-BIO and BIG trials (Appendix A), which only differ in the type of the applied anti-CD20 antibody—Ofatumumab and Obinutuzumab, respectively. At therapy start, patients in the CLL2-BIO trial in different response categories (BIO_IR/LR, BIO_NR) had significantly different CD20 expression levels compared to patients in the uMRD group of the same trial (IR/LR *p* = 0.0002; NR *p* = 0.00005), which was not observed for patients in the CLL2-BIG trial (Figure 7; DX). Despite high variations in the CD20 MFI (especially BIO_IR/LR), no significant differences were observed between the different time points RE, M3, and M5, nor between the different response groups in on-therapy samples.

## 4. Discussion

Flow cytometry can be used to evaluate the expression of a limited number of markers on different cell subsets in biological specimens. Short-term measurements can address the technical complexity of the method or variability of instruments to a great extent by an adequate experimental design and the use of controls. However, for longitudinal studies, which are part of clinical trials or long-term research projects, it is difficult to address factors causing variations in fluorescence intensities as environmental conditions and instrument settings can change considerably, even when reagent quality is maintained. The read-out of MFIs in standardized analytical settings as a direct measure of marker expression has been reported to provide concordant data with coefficients of variation for several markers [37], comparable to those obtained using calibration beads [38]. One can then propose that control of technical variability by standardized procedures should be able to tease out the biological variation observed between samples. As a member of the EuroFlow Consortium and ISO 15189 accredited lab, our laboratory applies the recommended EuroFlow standardization procedures in daily routines [15]. In contrast to the technical issues described by the authors, the variations observed on our instruments were all within the established quality control limits and were mainly not related to maintenance events or lot changes in the quality control beads. To avoid provoking differences in longitudinal measurements becoming significant by falling in measurement periods with quality controls near the upper or lower limits, a simple normalization procedure was designed and tested here. By using the ratio of median fluorescence intensities of the 7th peak of the daily measured quality control beads to its corresponding reference value for each channel and applying this factor to the other peaks, we were able to reduce the variability in the quality control measurements by about 2 to 10-fold in the four tested channels. More intriguingly, by balancing the fluorescence intensities using the calculated normalization factor, the observed higher MFI switches towards the upper and lower limits were almost completely eradicated throughout the fluorescence intensity measurement range, as confirmed by the other peaks of the quality control beads.

The 5 × 3 × 3 experiment to address variability on our flow cytometers confirmed that the high proportion individual donor samples contributed to the observed overall variability. The variability reported here is in concordance with previously published data [37,38,39,40], which also reported up to 30% or more CV based on molecules of equivalent soluble fluorochrome (MESF) or fluorescence intensities for markers on different cell subsets and different instruments. Mizrahi et al. [40] reported that the variability in marker expression could be reduced from ~20% to 3–15% by conversion of the MFI to MESF using calibration beads. Although adding to costs, the use of calibration beads might be an advisable approach for planned prospective studies; it cannot be used for retrospective analysis and might also not be best suited for studies within large clinical trials where samples come more or less frequently, often unannounced or on short notice. Only up to 10% of the measurement variability is related to the instrument; thus, a simple normalization approach from the daily quality control as presented here would reduce measurement variations caused by unknown factors influencing the instrument performance. Significance in the pairwise t-test did not change in the comparisons of normalized to not normalized data in the CD20 expression study from samples of the four CLL2 trials, supporting the strategy from the EuroFlow consortium aiming for combining multi-center, multi-instrument data. However, in contrast to the assumed low inter-individual variability discussed by Kalina et al. [37], our data suggest that once the standardized Euroflow procedures are used, the highest portion of variance is caused by inter-individual differences in marker expression, which exceeds the technical variance caused by the instrument itself as well as the standardized sample preparation and staining procedures. Despite this, the fraction of variance related to daily staining procedures, reagent lots, or operators account for ~30% in our experiments and still comprise a substantial part of the observed variations, which can only partially be influenced by following standardized staining procedures.

Finally, we investigated the CD20 expression in patients enrolled in four phase-II trials of the GCLLSG. In light of the availability of different treatment schemes for CLL and patients relapsing or being refractive to therapy [11,41], the question of whether CD20 expression might influence the outcome of patients in R/R therapy has only rarely been addressed by in vivo longitudinal studies. Additionally, only two long-term studies considering the expression of CD20 as the targeted B cell marker in such conditions have been published [10,42]. While in patients with non-Hodgkin lymphoma, a reduced CD20 expression at therapy start was linked to poor prognosis [43,44], reduced CD20 expression is a molecular feature of CLL cells [25,45,46]. In light of such findings, CD20 expression before treatment with anti-CD20 antibodies has been controversially discussed as a potential predictive marker for response to anti-CD20 therapy [27,28] or improved response in combination with the development of infusion-related reactions [47]. As expected from the studies of Boettcher et al. [10] and Skarczynski et al. [42], CD20 expression levels, though variable to some extent, do not rise under CD20 maintenance therapy in the CLL2-BXX trials. Although the resulting patient numbers in the different response subgroups are to some extent low, no significant difference in CD20 MFI was observed for patients in relapsed refractory treatment compared to patients in first-line treatment prior to therapy. Thus, the observed heterogeneity of CD20 expression might rather reflect the maturation status and/or underlying cytogenetic/molecular features of the malignant B-cells, as suggested by Maddy et al. [48], than the influence of an earlier anti-CD20 therapy.

However, our results suggest that the therapy outcome might be influenced by a combination of the anti-CD20 drug and the starting CD20 expression levels; for the therapy line including Ofatumumab (BIO), higher surface CD20 expression seems to be needed to reach a molecular MRD response leading to undetectable MRD, especially when compared to the identical regimen including Obinutuzumab (BIG). The BTK-inhibitor ibrutinib used in both therapy schemes has been shown to affect CD20 expression levels, thereby affecting the antitumor activities of the anti-CD20 drug [42,49]. The mode of the antitumor activity of both anti-CD20-directed therapies includes pathways for antibody-dependent cytotoxicity but differs for both agents. While Ofatumumab as a type I antibody acts via pathways of complement-dependent cytotoxicity, Obinutuzumab as a type II antibody affects the tumor cells by inducing direct cell death [50]. Compared to patients showing limited response at the MRD level, the patient group receiving Ofatumumab who showed deep molecular MRD response apparently benefits from the higher CD20 levels on the surface membrane of CLL cells. This might potentially be related to the higher frequency of trisomy 12-positive CLL cases in this group, as well as the mechanisms of the CD20 concentration-dependent antitumor activities of Ofatumumab. In line with our findings, a correlation between CD20 expression and complement-mediated lysis of CLL cells has been previously reported for the type-I antibody rituximab [51]. The highest in vitro bivalent binding capacity to Daudi cells, a B-cell line originating from Burkitt lymphoma, has been described for Ofatumumab in comparison to rituximab and Obinutuzumab, indeed resulting in lower binding dynamics, but also proposing a correlation with better CDC activation [52]. CDC activation in a BCR-dependent manner by Ofatumumab has also been reported in clinical CLL samples [53].

## 5. Conclusions

Technical variance in flow cytometric studies focused on evaluating marker expression levels can be at least partially corrected through the use of a normalization approach derived from the daily instrument quality control. This robust approach allows for the comparison of data even in long-term retrospective studies. It provides interesting insights into the correlation between CLL CD20 expression in the setting of different anti-CD20 targeted therapies and (MRD) responses to therapy. However, analysis of larger patient cohorts is still required to allow firm conclusions on the predictive value of CD20 expression in such studies.

## Figures and Tables

**Figure 1 cancers-14-04917-f001:**
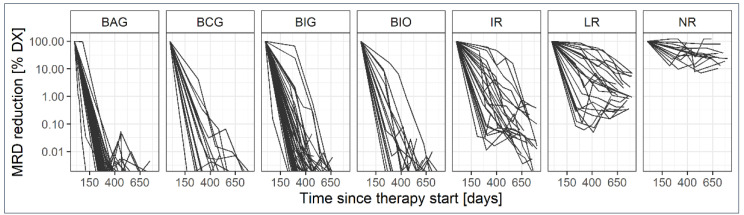
Distribution of patients of the CLL2-BXX trials among the defined MRD response groups (see Table 1 for scoring to MRD response groups); BCG, BAG, BIG, BIO: patients with deep (uMRD) MRD response under the respective trial schemes (B: Bendamustine; I: Ibrutinib; A: Venetoclax; C: Idelalisib; G: Obinutuzumab; O: Ofatumumab), IR: intermediate MRD response; LR: limited MRD response; NR: Non-Responders; IR/LR/NR include patients from the different trial schemes.

**Figure 2 cancers-14-04917-f002:**
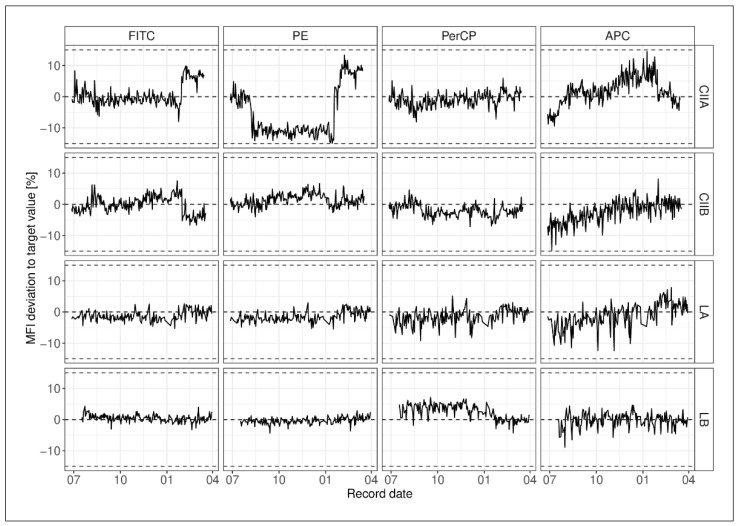
Variability of daily quality control measurements of peak-7 Rainbow beads’ MFI differences to the target values on different flow cytometry instruments (CIIA, CIIB: CantoII; LA, LB: Lyric) under equal environmental conditions during a time period of 8 months.

**Figure 3 cancers-14-04917-f003:**
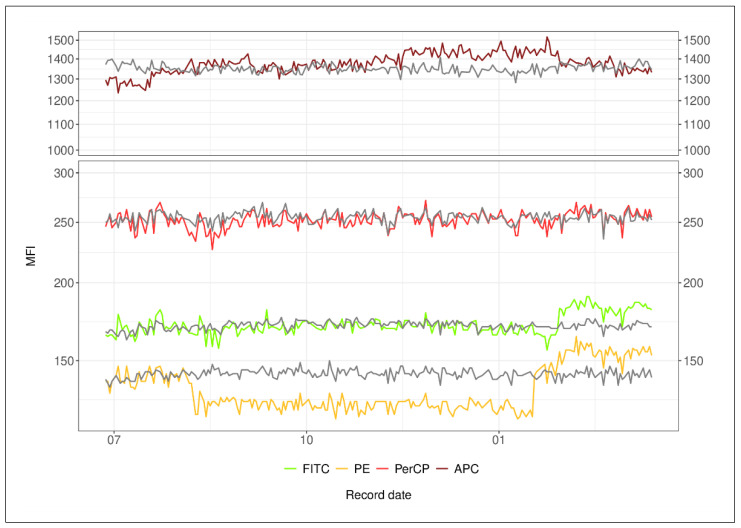
Measured (colored) and normalized (grey) MFIs of peak 2 of the Rainbow beads for instrument CantoIIA during an 8-month time period (July 2018 to April 2019).

**Figure 4 cancers-14-04917-f004:**
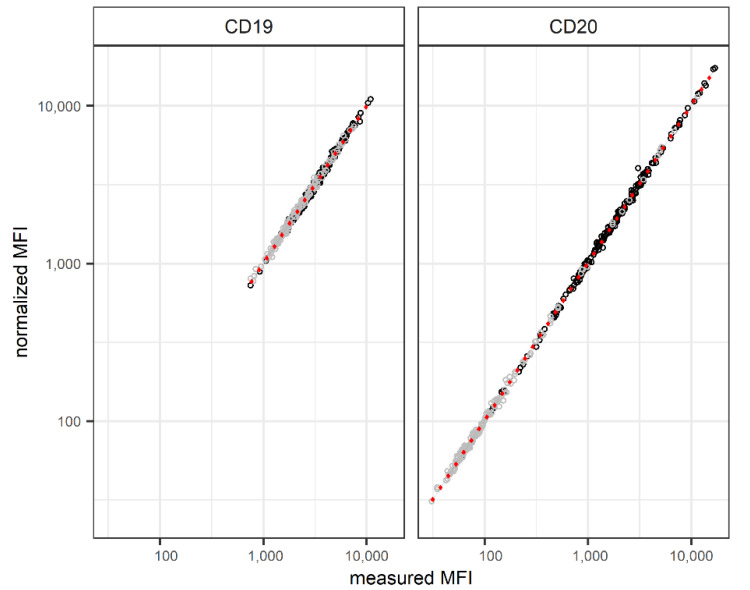
Correlation plot of measured and normalized MFIs for CD19-PerCP Cy5.5 and CD20-FITC in 196 samples prior to therapy (black) and 166 on-therapy (grey) samples from 196 patients enrolled in the CLL2-BXX trials. Linear regression (red) is close to 1 (R^2^ CD19: 0.997; R^2^ CD20: 0.999), as NF used for normalization was between 0.9 and 1.13.

**Figure 5 cancers-14-04917-f005:**
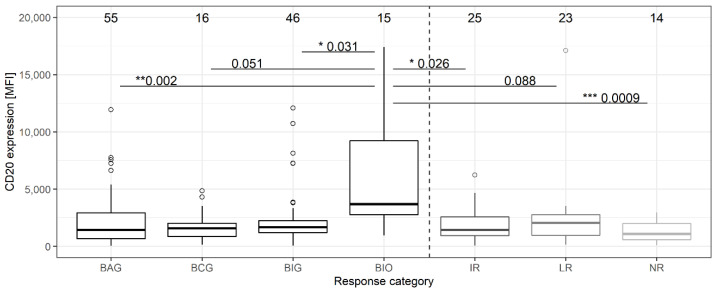
CD20 expression on CLL cells of patients in the MRD response groups prior to therapy start. Number of patients in each group is given above. Reported *p*-values for group differences were obtained from Dunn’s test. BCG, BAG, BIG, BIO: patients with deep (uMRD) MRD response under the respective trial schemes (B: Bendamustine; I: Ibrutinib; A: Venetoclax; C: Idelalisib; G: Obinutuzumab; O: Ofatumumab), IR: intermediate MRD response; LR: limited MRD response; NR: Non-Responders; IR/LR/NR include patients from the different trial schemes. * *p* ≤ 0.05; ** *p* ≤ 0.01; *** *p* ≤ 0.001.

**Figure 6 cancers-14-04917-f006:**
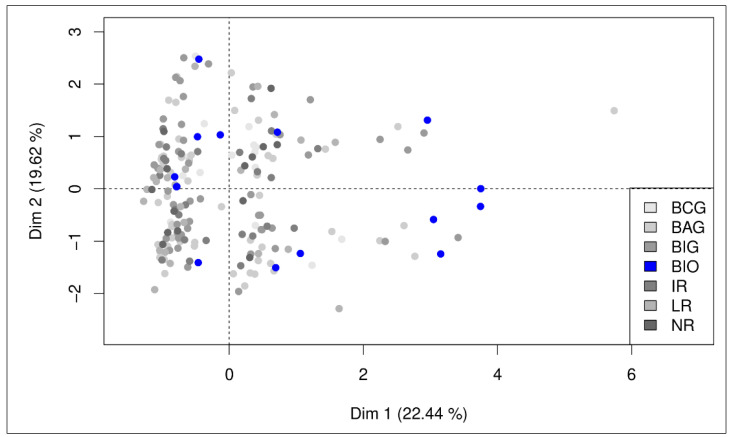
Observation map of mixed principal component analysis of pre-treatment samples using age and CD20 expression levels as continuous variables and Binet stage, *IGHV* mutational status, trisomy 12, and therapy strata as categorical variables.

**Figure 7 cancers-14-04917-f007:**
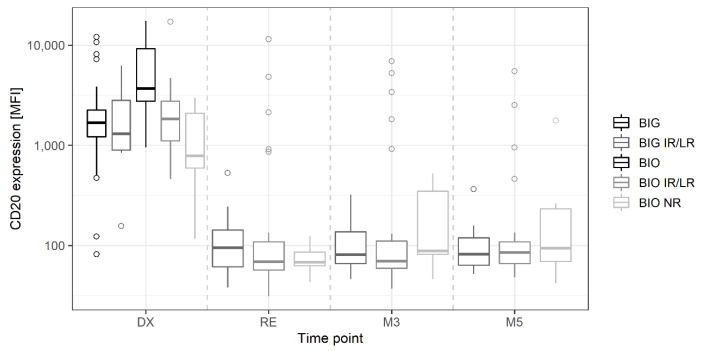
Expression of CD20 (MFI) on CLL cells in diagnostic (DX) and on-therapy samples (final restaging: RE; maintenance cycle 3 or 5: M3, M5) for patients in the CLL2-BIO and CLL2-BIG trials, scored according to their MRD response. MRD response categories include patients with medium and weak responses according to the therapy scheme (BIG_IR/LR, BIO_IR/LR), the non-responders in the BIO trial (BIO_NR), and patients with uMRD response (BIG, BIO).

**Table 1 cancers-14-04917-t001:** Criteria for MRD response scoring to study CD20 expression.

Patient Groups Based on Molecular Response up to 1.5 Years of Maintenance Therapy in the Different Trial Schemes				
	Deep Responder (uMRD)	Intermediate Responder (IR)	Limited Responder (LR)	Non-Responder (NR)
Sustained undetectable MRD (<10^−4^, uMRD) before/during 1.5 years maintenance therapy	Yes	No	No	No
MRD reduction compared to timepoint before treatment	>4 LOG	2–4 LOG	>1–3 LOG	≤1 LOG
Increasing MRD under maintenance therapy	Yes, patient in low-risk group MRD < 10^−4^	Yes, Moderate, patients in medium-risk group MRD ≥ 10^−4^; <10^−2^	Yes, faster MRD increase, patients in high-risk group MRD ≥ 10^−2^	Patients in high-risk group MRD ≥ 10^−2^

**Table 2 cancers-14-04917-t002:** Variability of flow cytometric expression analysis based on MFI measurements using PE-conjugated antibodies targeting different markers on cell populations of PB of healthy donors. Coefficients of variation (CV) and their influence on the observed variance based on the effect size (η^2^) were determined by ANOVA analysis.

Marker	Grand Mean MFI	Donor		Instrument		Day	
		CV [%]	η^2^	CV [%]	η^2^	CV [%]	η^2^
CD22	7090	37.29	0.792	7.19	0.015	15.76	0.071
CD38	1758	115.7	0.954	6.36	0.001	9.07	0.003
CD56	651	26.71	0.263	9.37	0.016	42.21	0.328
CD79b	5805	88.50	0.670	8.91	0.003	62.97	0.170

## Data Availability

The data presented in this study are available in this article (and Appendix A).

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
