# Peer review of "CD20 Expression as a Possible Novel Prognostic Marker in CLL: Application of EuroFlow Standardization Technique and Normalization Procedures in Flow Cytometric Expression Analysis"

_cancers, 2022, doi:10.3390/cancers14194917_

Round 1

Reviewer 1 Report

The paper is well written and would interests scientists and medical community focused on flow cytometry. Indeed, it deals with a recurrent issue which deserves technical description: the difficulties to obtain and to normalize median fluorescence intensities (MFI), especially for longitudinal purposes. Methods are clear and a major standardization effort has been made by this team. The article is well constructed and easy to understand.

However, the article is surprisingly focused on normalization approaches. Maybe, a journal focused on methods would be more relevant.

Author Response

Dear Reviewer,

thank you very much for your kind comments on our manuscript.

The role of CD20 expression level on response and clinical prognosis in CLL is still a matter of debate. To address this question also in the context of two different generation antibodies (Ofatumumab vs Obinutuzumab) we analyzed data acquired in the CLL2-BXX trials. To identify bias due to technical influences in the measurements of CD20 expression we investigated the influence of seasonal trends, different flow cytometers and other factors in longitudinal data. This is summarized in the technical results section (3.1 and 3.2). We feel, that the Section of Methods and Technologies Development of the Journals Special Issue on Leukemia and Lymphoma immunophenotyping is well suited for our manuscript addressing technical questions of immunophenotyping results in the context of the ongoing CD20 expression vs patient’s genetic features vs treatment dispute. As shown in the manuscript, the standard EuroFlow procedure is feasible to overcome external influences in expression analysis based on immunophenotyping, and no further normalization process is needed.

Kind regards

Anke Schilhabel and Matthias Ritgen

Reviewer 2 Report

The paper provides interesting information

I suggest

1. to include the numebr of patients studied at the beginnig of the RESULT SECTION, along with the number of treatment naive (TN) patients and rel/ref (R/R) patients.

2. To report and comment whether any difference was observed in CD20 expression in TN and R/R patients.   

Author Response

Dear Reviewer,

thank you very much for your comments to our manuscript. We feel that your comment was extremely helpful in strengthening the manuscript results section by better embedding the technical question of data normalization in the clinically addressed question.

We tried to improve the description of Material and Methods section as said by including the patient numbers of the selection process. Additionally, the summary of the patient characteristics in the Supplementary Material is already referenced at this point to allow readers a comparison against the clinical cohorts. The criterion for MRD detectability in Table 1 was refined to “sustained undetectable MRD” to highlight the differences between the MRD response groups. The experiments using the Quantibrite-PE beads to address the measurement variability in flow cytometric analysis was added.

To point 1:

The results section was re-structured to start with the description of patient numbers and selection, including the requested number of treatment naïve and relapsed/refractory patients.

To point 2:

We included a Figure S2 on CD20 expression results in both of these sub-groups in the Supplementary material. To further describe CD20 expression in the context of the MRD response groups and treatment schemes as well as treatment line,  detailed values  are presented in the Supplementary Material Table S3. The results for these subgroups had already been discussed in the submitted version of the manuscript. However, the comment on this in the Discussion section is now adapted to include weakening of statistical statements by the resulting low patient numbers when patients in the different response groups are further grouped according to the treatment line.  

Kind regards

Anke Schilhabel and Matthias Ritgen

Round 2

Reviewer 1 Report

Some sections have been greately improved, by providing additional information. As previously mentioned, this article is focused on technical issues, and the author response, arguing that this article is dedicated to the Section of Methods and Technologies Development of the Journal Special Issue is appropriate. Considering that, I think that this article perfectly fit with the scope of the issue.

The article is well-builded and of interest in the field of intensity analyses.